# Antibiotic Use and Antibiotic Resistance: Public Awareness Survey in the Republic of Cyprus

**DOI:** 10.3390/antibiotics9110759

**Published:** 2020-10-30

**Authors:** Mikaela Michaelidou, Spyridon A. Karageorgos, Constantinos Tsioutis

**Affiliations:** 1School of Medicine, European University Cyprus, 2404 Nicosia, Cyprus; mm152237@students.euc.ac.cy (M.M.); s.karageorgos@external.euc.ac.cy (S.A.K.); 2Department of Pediatrics, Limassol General Hospital, 4153 Kato Polemidia, Cyprus

**Keywords:** antibiotic resistance, antibiotic-resistant bacteria, antibiotic use, inappropriate, interventions, campaigns, awareness

## Abstract

We aimed to assess the knowledge and understanding of antibiotic use and resistance in the general population of Cyprus, in order to inform future antibiotic awareness campaigns with local evidence. Cross-sectional survey following the methodology of the “Antibiotic resistance: Multi-country public awareness survey” of the World Health Organization, during December 2019–January 2020. A total of 614 respondents participated: 64.3% were female and most were aged 35–44 years (33.2%) or 25–34 years (31.8%). One-third had used antibiotics >1 year ago and 91.6% reported receiving advice on appropriate use from a medical professional. Despite high awareness on correct use of antibiotics, lack of knowledge was noted for specific indications, where approximately one-third believed that viral infections respond to antibiotics and 70.7% lack understanding of how antibiotic resistance develops. Higher education graduates exhibited significantly higher knowledge rates. As high as 72.3% were informed about “antibiotic resistant bacteria” from healthcare professionals or social media. Most agreed on the usefulness of most suggested actions to address antibiotic resistance, with higher proportions acknowledging the role of prescribers. Up to 47% could not identify their role in decreasing antibiotic resistance. Our study provides local evidence to inform future efforts in a country characterized by high antibiotic consumption rates.

## 1. Introduction

Recent and inappropriate use of antibiotics are well established risk factors for infection with resistant bacteria [1,2]. Consequently, promoting prudent antibiotic use should represent the cornerstone of strategies to prevent and control the emergence and spread of resistant bacteria. Among these, educating patients and healthcare professionals, as well as enhancing communication between them, are important steps to decrease resistance by increasing knowledge on appropriate antibiotic use [3]. However, to be able to implement effective education, thorough understanding of patients’ level of knowledge, understanding, and attitudes regarding antibiotic resistance and antibiotic use is needed [4].

In 2018, the European Commission conducted a repetitive Eurobarometer survey on the knowledge, attitudes and behavior of the public towards antibiotic use and after having developed guidelines on sensible use of antimicrobials in human medicine in 2017 [5]. Although improvements were noted compared to the previous surveys, certain areas render further investigation to identify knowledge and behavioral gaps among the general population that might lead to misuse and overprescribing of antibiotics [5,6]. For example, among all respondents in the Eurobarometer study, 14% and 12% reported that they received antibiotics for sore throat and the flu respectively [5], whereas similarly, 12% of respondents in a 2017 multinational survey self-managed sore throat with antibiotics [7].

In Cyprus, results from local surveillance of antimicrobial use and resistance are limited to international studies. According to recent surveillance reports [8], Cyprus is among the highest consumers of antibiotics in the community and hospitals in Europe, with extended spectrum antibiotics (e.g., third and fourth generation cephalosporins, fluoroquinolones, carbapenems) accounting for more than 50% of antibiotic use in hospitals [9]. As high as 95% of surgical prophylaxis is administered for more than 1 day in Cyprus, reflecting the degree of inappropriate antibiotic use. Antibiotic-resistant bacteria are isolated in up to 51.4% of healthcare-associated infections in Cyprus, which is exceeded by only six countries in the European region and is alarmingly high compared to the European average of 31.6% [10]. In addition, previous surveys (e.g., the Eurobarometer survey) have shown that Cyprus citizens have exhibited certain knowledge gaps in regards to indications for antibiotic treatment, such as the common cold and the flu [5]. This is further supported by a previous survey among parents in Cyprus, where lower education level was associated with antibiotic misuse in their children [11].

The purpose of the current study is to investigate in detail the level of understanding, knowledge and common behaviors related to antibiotic use and antibiotic resistance in the general public of Cyprus, aiming to guide future efforts to improve the effectiveness of awareness campaigns and enhance optimal measures to improve antibiotic use in the community.

## 2. Results

### 2.1. Participants and Demographics

During the study period, a total of 614 respondents completed the survey (64% females) and the majority were between the ages 25–44 (65%). Among responses, 283 (46.1%) were completed during face-to-face interviews and 331 (53.9%) were completed online. Most were residents of urban (63.4%) or suburban areas (22.8%) and 13.8% lived in rural areas. The majority of participants were Nicosia residents (49%), followed by residents of Limassol (19%), Larnaca (17%), Famagusta (8%), and Paphos (7%). University/college graduates and master’s degree holders represented 79% of participants.

Concerning household composition, most were married adults, either with at least one child (24.8%) or without children (21.7%). Household with one adult was also common (31.1%), with less frequent being multiple adults (10.3%) or multiple adults with at least one child under 16 (6.5%). Details on demographics of participants are presented in Table 1.

### 2.2. Use of Antibiotics

A total of 194 of responders (32%) received their last course of antibiotics within 6 months and the majority (87%) reported they received their prescription from a doctor. As high as 91.6% received advice on how to take their course of antibiotics by either a doctor, a nurse or a pharmacist. Additionally, 96.3% of responders took their antibiotics from a pharmacy store. No differences were detected in recent antibiotic use when comparing gender, level of education, and urbanization between respondents.

### 2.3. Knowledge on Appropriate Antibiotic Use

Among respondents, 87% stated that they know they need to take the full prescription of antibiotics and 79.6% stated that it is not acceptable to use the same antibiotics taken from a family member or friend with the same symptoms.

Misconception was recorded when respondents were shown the statement “It’s acceptable to buy the same antibiotics, or request these from a doctor, if you’re sick and they helped you get better when you had the same symptoms before”, where 25.4% incorrectly agreed.

When participants were asked to select which medical conditions can be treated with antibiotics, 81% and 61.9% correctly identified bladder/UTI and skin/wound infections, respectively. In contrast, 36.6% and 30.0% incorrectly chose the common cold/flu and sore throat respectively, as conditions that require antibiotic treatment. Indication for treatment of gonorrhea with antibiotics was correctly selected by only 23.1% (Figure 1).

No differences were detected regarding knowledge on appropriate antibiotic use between gender, urbanization, level of education, and household composition.

### 2.4. Antibiotic Resistance Understanding and Awareness

Concerning basic knowledge of antibiotic resistance terms, 21.2% of respondents stated that they had never heard any of the terms: “antibiotic resistance”, “antimicrobial resistance”, or “antibiotic-resistant bacteria”. The phrase known best by 72.3% of respondents was “antibiotic-resistant bacteria”, followed by “antibiotic resistance” (55.6%) and “antimicrobial resistance” (33.7%). Among respondents who had heard these terms before, approximately one-third stated they had heard them from a doctor/nurse or the social media.

To explore the level of understanding of antibiotic resistance, respondents were presented a list of statements and asked whether these were true or false, or their level of agreement. Overall, the majority of respondents correctly identified most statements about antibiotic resistance and how to address it. A frequent misconception was detected regarding the statement that the human body becomes resistant to the antibiotics and not the bacteria themselves, where 70.7% incorrectly identified it as a true statement. Knowledge-based questions were answered correctly to a higher level by tertiary education graduates compared to secondary education graduates (*p* < 0.001) and by residents of urban regions compared to suburban and rural residents (*p* = 0.024) (Table 2). In addition, adults with children were more aware that resistant bacteria can be spread from one person to another (*p* = 0.005), whereas they wrongly believe that only people that use antibiotics regularly are at risk of antibiotic resistance (*p* = 0.019).

Regarding perceptions towards antimicrobial resistance, participants exhibited a high level of agreement in the majority of statements (Figure 2). On the contrary, only 62.7% and 58.6% of the respondents agreed that pharmaceutical companies should develop new antibiotics and that governments should reward the development of new antibiotics, respectively. No significant differences were detected between demographic groups.

Participants in their majority agreed that everyone needs to use antibiotics responsibly (92.8%) and are worried on the impact this problem will pose on their health and family (76.4%). Misconceptions were identified when they were asked about the impact of antibiotic resistance and about their role in addressing this issue. Specifically, 52.3% believe that medical experts will solve the problem of antibiotic resistance before it becomes too serious and as low as 63.4% include antibiotic resistance among the biggest problems the world faces. Respondents also believe that if they use antibiotics correctly, antibiotic resistance will not be an issue for them (47.6%), and that there is not much that people can do to stop antibiotic resistance (42.5%) (Figure 3). Questions regarding attitudes towards antibiotic resistance were answered correctly to a higher level by tertiary education graduates and adults with children (Table 3).

## 3. Discussion

Recent surveillance reports highlight that Cyprus has a high prevalence of antibiotic-resistant bacteria, as well as a high overall consumption of antibiotics [9,10,12]. The current survey provides essential information on knowledge and understanding of the public of Cyprus on use of antibiotics and antibiotic resistance and delineates areas for future interventions to improve antibiotic use and decrease inappropriate use. Specifically, the results of this survey show that even though the general public of Cyprus have basic knowledge and exhibit awareness on appropriate antibiotic use and antibiotic resistance, they do not fully understand basic premises, such as the causes of antibiotic resistance or their own role in decreasing it. Certain knowledge gaps and misconceptions were detected regarding mechanisms leading to antibiotic resistance, which diseases are indicated for antibiotic treatment and the need for a multifaceted approach to tackle resistance [13]. Consequently, specific targets are set for improvement of knowledge and awareness on the mechanisms leading to antibiotic resistance, as well as which diseases can be treated with antibiotics and which cannot. Moreover, our study has highlighted the importance of enabling public understanding of their own role in reducing antibiotic resistance and the need for a multidisciplinary approach to tackle resistance [14].

Optimization of antibiotic use in healthcare requires a national strategy with specific priorities and with participation of different stakeholders involved in antibiotic use, such as prescribers, pharmacists, patients, and healthcare organizations. A minimum set of knowledge and competences in microbiology, infection pathogenesis, prescribing, and stewardship, is necessary for prescribers in order to ensure safe and responsible antibiotic use [15]. In addition to organized antimicrobial stewardship and surveillance programs, interventions that have shown to improve antibiotic use-related indices have included setting standards and quality indicators for antibiotic use, and supporting doctor-patient communication [16]. Especially for the latter, training physicians in communication skills and encouraging shared decision making when prescribing antibiotics to a patient, have been repeatedly shown to improve the quality of antibiotic use without affecting outcomes [17,18].

Compared to the corresponding World Health Organization (WHO) multi-country survey [19], Cypriots showed higher rates of knowledge about the appropriate use of antibiotics. For instance, incorrect answers to the statements “it is acceptable to buy the same antibiotics, or request these from a doctor if you are sick and they helped you get better when you had the same symptoms before” and “it is okay to use antibiotics that were given to a friend or family member, as long as they were used to treat the same illness”, were 25.4% and 20.4% respectively in our survey, compared to the respective WHO average of 43% and 25% [19]. However, these findings still suggest that one out of four people are inclined to use antibiotics that helped them or others in the past for the same disease. This is of major concern, since prescription based on clinical reasoning is the cornerstone of appropriate antibiotic use [20]. The present survey was performed 6 months after the establishment of the General Health System in Cyprus, which mandates antibiotic use only through electronic prescription and over-the-counter antibiotics are not allowed. Although 50% of participants reported having received their last antibiotic course at least a year ago (Table 1), 87% reported using a prescription. Therefore, the proportion of respondents that could have used antibiotics over-the-counter is low. In this case, it is the role of healthcare professionals to focus on educating patients regarding indications of antibiotic use.

In line with the findings of previous surveys that followed the same methodology [19,21,22,23], certain misconceptions were identified regarding the problem of antibiotic resistance. For example, the majority of respondents (70.7%) falsely believe that antibiotic resistance occurs when their body becomes resistant to antibiotics. Half of respondents also falsely believe that “antibiotic resistance is only a problem for people who take antibiotics regularly” and 42.5% believe there is not much they can do to stop antibiotic resistance. Hence, despite basic awareness of these terms, future campaigns should focus on education on how resistance develops, how resistant bacteria are spread, as well as the role of each one in mitigating resistance. Such misperceptions have been recorded in a recent systematic review, where the majority of survey participants falsely attributed antibiotic resistance as a change in the human body [24]. Furthermore, only 36% of participants recorded in the above systematic review mentioned having discussed the issue of antibiotic resistant with their doctor. To this end, when addressing doctor–patient communication, future campaigns should enable the role and engage of the general public in addressing antibiotic resistance through shared decision making [18,24,25].

Notably, tertiary education graduates exhibited significantly higher levels of understanding of such statements, further supporting the role of education in improving understanding of basic antibiotic resistance concepts. This is also confirmed by previous studies that included university students from various countries, who showed high levels of knowledge [23,26,27]. Importantly, even in these cohorts in our study, certain misconceptions were recorded, such as sharing of antibiotics, use of previous antibiotic regimens, effectiveness of antibiotics in specific indications, and role in addressing the antibiotic resistance issue. Although these findings set specific targets to improve understanding from an undergraduate level, there is need to differentiate whether health or life sciences students perform better than students from other study programs.

Αdults with children displayed higher level of understanding in statements assessing attitudes towards antibiotic resistance. This is an interesting finding since parental knowledge and misconceptions have been identified as major contributors for unnecessary antibiotic use in children [28]. The increased level of parental knowledge is in line with findings from a previous study assessing parental knowledge on antibiotic use in children with upper respiratory tract infections in Cyprus [11]. These findings combined, support that parents with children are more aware of the problem of antibiotic resistance and misuse and thus, adults without children and adult medical specialties could represent the primary target of future interventions towards improving awareness and antibiotic use in Cyprus.

The majority of respondents in our study agreed that almost all proposed actions could help address the problem of antibiotic resistance, particularly for statements signifying the importance of receiving antibiotics only when prescribed by healthcare professionals. This again is in agreement with other observations of our survey and the Eurobarometer findings, showing that a significant proportion of respondents take antibiotics only by prescription [5]. In contrast, the statements “governments should reward the development of new antibiotics” and “pharmaceutical companies should develop new antibiotics” were the least commonly agreed to, demonstrating lack of comprehension of a multidisciplinary approach to combatting antimicrobial resistance and developing new antibiotics (Table 3). On the other hand, this could also be due to disbelief towards the true incentives of the government and industry in regards to public health priorities.

The results of our study should be interpreted in the context of certain limitations. Some information and sampling bias might be owed to geographic distribution, due to low percentage of rural residents (13.8%), which was due to lower accessibility to the survey. Despite the fact that responses were collected from all government-controlled districts, slightly higher proportion of responses from Nicosia and lower from Limassol were noted, compared to the population of each district [29]. Geographic distribution could in turn have affected representation of the education level of participants, as most respondents were higher education graduates (80.1%). Admittedly, the latter finding is higher compared to the current proportion of higher education attainment in Cyprus, albeit among the highest in Europe (49.5% for those aged 25–54 years and 23.5% for those aged 55+ years in 2019) [30]. This further enforces the necessity of proposed interventions, as we have shown association between education level and knowledge and understanding.

As for population representativeness, compared to the previous studies that followed the WHO methodology, the number of respondents is considered far more representative of the total population of the Republic of Cyprus (70 respondents/100,000 population) [19,21,22,26,29]. There was sufficient participation from different age groups, with those in young adulthood (25–44 years of age) representing 65% of the population surveyed and corresponding well to the age distribution of the Cyprus population [29]. The responses of the current survey were gathered from a convenience sample, including results from both online and face-to-face responses, might have led to response bias, where people answer questions on the basis of what is ‘expected’ of them to answer. However, approximately 50% completed the survey during face-to-face interviews and therefore this type of bias was reduced. The survey also contained several statements asking the respondents whether they agreed. This could lead to acquiescence bias, where respondents are more likely to agree rather than disagree, regardless of the content. Nevertheless, this survey followed an established and validated methodology by the WHO, with important findings from a notable sample size of the general public of Cyprus, a country characterized by high antimicrobial use and alarming antimicrobial resistance rates.

## 4. Materials and Methods

### 4.1. Questionnaire Adaptation and Development

The present study adapted the methodology of the WHO “Antibiotic Resistance: Multi-Country Public Awareness Survey” [19]. This survey consists of a questionnaire completed in 2015 by nearly 10,000 respondents in 12 countries from all WHO Regions (Barbados, China, Egypt, Nigeria, India, Indonesia, Mexico, Russian Federation, Serbia, South Africa, Sudan, Vietnam). The survey questionnaire was subsequently adapted by other countries and applied in the general public in Italy [21], and Nigeria [22], in pharmacy students in Australia and Sri Lanka [26], and in undergraduate students in Brunei [23]. This questionnaire was selected for the present study, because it was developed and applied over a broad number of persons by the WHO and also because our results will allow for comparisons with findings from other countries. The questionnaire and detailed study methodology are available online (https://apps.who.int/iris/handle/10665/194460).

Following permission for use by the WHO and approval by the Cyprus National Bioethics Committee (no. 2019.01.178), the questionnaire was developed in Greek, back-translated, pilot-tested with five persons and adjusted appropriately. It was available on paper for face-to-face interviews or online as a prepared Google^®^ forms survey (Appendix A).

The questionnaire comprised of 7 multiple-choice questions requesting: gender, age, city of residence, urbanization, education, income and household composition and 35 questions regarding knowledge and perceptions about antibiotic use and antibiotic resistance. Of those 35 questions, 11 were multiple-choice, 10 were true/false, and 14 were Likert scale questions.

### 4.2. Study Setting, Duration, and Participants

The study was performed from 21st December 2019 until 31st January 2020. Respondents from around Cyprus were invited to complete the survey in public areas such as cafeterias, shopping malls and other, where the primary study researcher approached the participants and after explaining the purpose of the study, they agreed on completing it. The survey was also available online as a Google^®^ forms document, which was sent personally via social media after the purpose was again explained. A reminder for completion was sent two days later. All questionnaires given or sent were answered and all questionnaires were included regardless of degree of completion. Questionnaires were given in all five districts of the government-controlled areas of the Republic of Cyprus: Nicosia (39% of total population), Limassol (28.0% of total population), Larnaca (16.8% of total population), Paphos (10.7% of total population), and Famagusta (5.5% of total population) [29].

Participation was voluntary, anonymous and without compensation. Informed consent was stated on the first page. To minimize methodological bias, participants were asked to self-complete the survey.

Participants were eligible to complete the survey if they were:Cyprus residentsAdults > 16 years’ oldNon-healthcare professionals

### 4.3. Statistical Analysis

Descriptive statistics were used for quantitative parameters. Unanswered questions were included in the results as ‘not answered’. Data are presented as counts (%) for categorical variables. Potential associations of knowledge, attitudes and perceptions with participants’ gender, level of education, urbanization, and household composition were assessed using the Chi-square test or Fisher’s exact test. Statistical significance was set at 5%. Data were processed and analyzed using GraphPad Prism version 8.0 for Windows (GraphPad Software, Inc., San Diego, CA, USA).

## 5. Conclusions

To our knowledge, this is the first survey to specifically assess knowledge, understanding and attitudes of the general public in Cyprus towards antibiotic use and antimicrobial resistance. Despite a high degree of understanding, areas for improvement were detected, including knowledge of indications of antibiotic use, mechanisms leading to antibiotic resistance, and the need of a multifaceted approach to combat resistance. These results will inform future engagement efforts to ensure data-driven campaigns and community stewardship programs, aiming to improvement of the quality of healthcare in a country characterized by high antibiotic consumption rates.

## Figures and Tables

**Figure 1 antibiotics-09-00759-f001:**
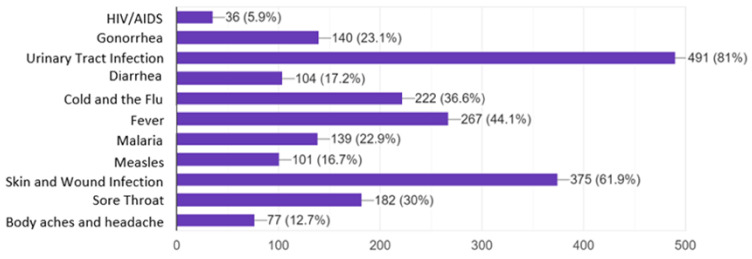
Level of knowledge on conditions that can be treated with antibiotics (*x*-axis shows number of responses).

**Figure 2 antibiotics-09-00759-f002:**
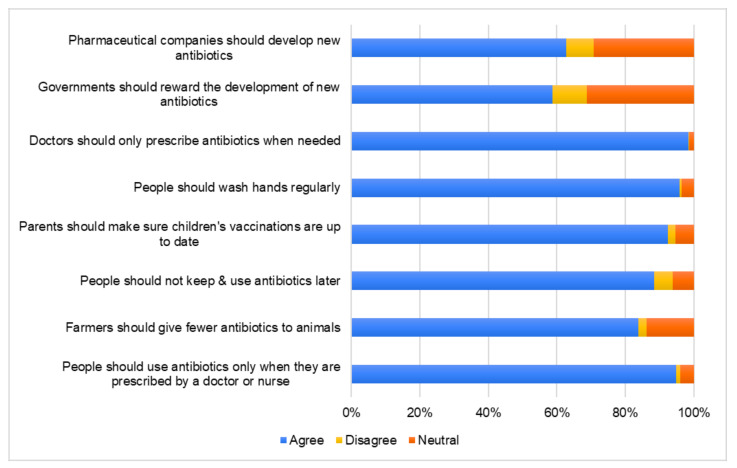
Perceptions of respondents towards antibiotic use and antibiotic resistance.

**Figure 3 antibiotics-09-00759-f003:**
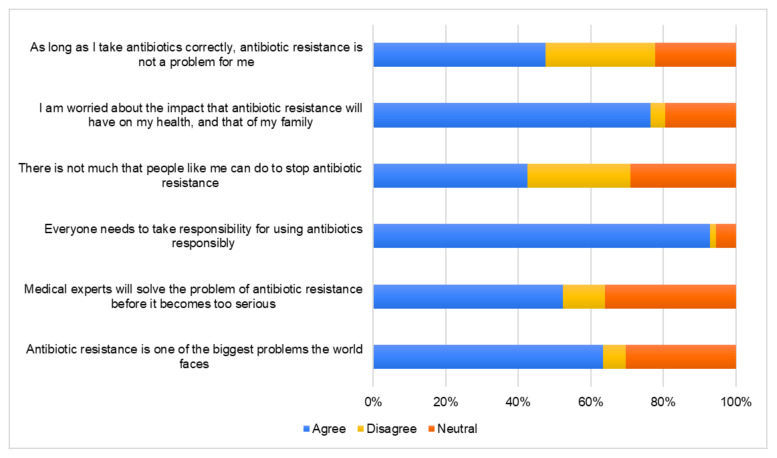
Percentage of responses from all respondents to statements surrounding attitudes toward antibiotic resistance.

**Table 1 antibiotics-09-00759-t001:** Socio-demographic characteristics of participants

Variable	Frequency *n* (%) ^a^
**Gender**	
Male	219 (36)
Female	395 (64)
**Age**	
16–18	6 (1)
18–24	62 (10)
25–34	195 (32)
35–44	204 (33)
45–54	78 (13)
55–64	57 (9)
65+	12 (2)
**District**	
Nicosia	303 (49)
Limassol	118 (19)
Larnaca	102 (17)
Famagusta	51 (8)
Paphos	40 (7)
**Level of education completed**	
No school	2 (0.3)
Primary	1 (0.2)
Secondary	7 (1.1)
High school	96 (16)
College/university	282 (46)
Master’s degree	203 (33)
Doctorate degree	15 (2.4)
Not answered	8 (1)
**Last antibiotic use**	
Last month	86 (14)
Last 6 months	108 (18)
Last year	90 (15)
More than a year ago	217 (35)
Never	25 (4)
Can’t remember	88 (14)
**Received prescription for last antibiotic use ^b^**	
Yes	515 (87)
No	39 (7)
Not remember	25 (4)
Not answered	10 (2)

^a^*n* = 614, ^b^
*n* = 589.

**Table 2 antibiotics-09-00759-t002:** Level of understanding of antibiotic resistance

		Education Level			Residence				Household Composition			
Statement ^a^	Correct Answer	Secondary *n* (%)	Tertiary *n* (%)	*p*-Value	Urban *n* (%)	Suburban *n* (%)	Rural *n* (%)	*p*-Value	Adults *n* (%)	Adults with Children Under 16 *n* (%)	Other *n* (%)	*p*-Value
Antibiotic resistance occurs when your body becomes resistant to antibiotics and they no longer work as well	F	31 (29)	147 (29)	1.000	127 (33)	35 (25)	17 (20)	**0.027 ***	91 (28)	73 (32)	15 (24)	0.360
Many infections are becoming increasingly resistant to treatment by antibiotics	T	87 (82)	434 (87)	0.215	326 (84)	125 (89)	76 (89)	0.213	275 (85)	197 (87)	57 (90)	0.471
If bacteria are resistant to antibiotics, it can be very difficult or impossible to treat the infections they cause	T	87 (82)	431 (87)	0.215	328 (85)	125 (89)	70 (82)	0.235	276 (85)	194 (85)	58 (92)	0.340
Antibiotic resistance is an issue that could affect me or my family	T	68 (64)	381 (77)	0.010 *	284 (74)	110 (79)	59 (69)	0.243	232 (72)	178 (78)	47 (75)	0.196
Antibiotic resistance is an issue in other countries but not here	F	89 (84)	458 (92)	0.029 *	354 (91)	126 (90)	74 (87)	0.500	286 (88)	210 (93)	58 (92)	0.224
Antibiotic resistance is only a problem for people who take antibiotics regularly	F	34 (32)	264 (53)	<0.001 *	205 (53)	64 (46)	32 (38)	0.024 *	144 (44)	128 (56)	29 (46)	0.019 *
Bacteria which are resistant to antibiotics can be spread from person to person	T	66 (62)	334 (67)	0.366	254 (66)	98 (71)	53 (62)	0.413	199 (61)	169 (74)	40 (63)	0.005 *
Antibiotic-resistant infections could make medical procedures like surgery, organ transplants, and cancer treatment much more dangerous	T	60 (57)	352 (71)	0.006 *	255 (66)	109 (78)	53 (62)	0.012 *	211 (65)	165 (73)	45 (71)	0.148

^a^ Presented as *n* (%) of respondents who answered question correctly. F: False; T: True. * *p*-value < 0.05.

**Table 3 antibiotics-09-00759-t003:** Participants’ (*n* = 614) attitudes toward antibiotic resistance

Attitudes Toward Antibiotic Resistance		Gender			Educational Level			Urbanization				Household Composition			
		Female	Male	*p*-Value	Secondary	Tertiary	*p*-Value	Urban	Suburban	Rural	*p*-Value	Adults	Adults with Children	Other	*p*-Value
Antibiotic resistance is one of the biggest problems the world faces	Agree	250 (63)	139 (63)	0.998	71 (67)	312 (62)	0.673	246 (63)	92 (66)	51 (60)	0.599	175 (54)	168 (74)	46 (73)	**<0.001 ***
	Neutral	120 (30)	66 (30)		29 (27)	155 (31)		115 (30)	43 (31)	28 (33)		126 (39)	47 (21)	13 (21)	
	Disagree	25 (6)	14 (6)		6(6)	33 (7)		28 (7)	5 (4)	6 (7)		23 (7)	12 (5)	4 (6)	
Medical experts will solve the problem of antibiotic resistance before it becomes too serious	Agree	215 (48)	106 (54)	0.293	69 (65)	247 (49)	0.005 *	206 (53)	72 (51)	43 (51)	0.649	167 (52)	118 (52)	36 (57)	0.561
	Neutral	138 (38)	83 (35)		32 (30)	187 (37)		142 (37)	51 (36)	28 (33)		124 (38)	78 (34)	19 (30)	
	Disagree	42 (14)	30 (11)		5 (5)	66 (13)		41 (11)	17 (12)	14 (16)		33 (10)	31 (14)	8 (13)	
Everyone needs to take responsibility for using antibiotics responsibly	Agree	373 (90)	197 (94)	0.066	98 (92)	464 (93)	0.992	365 (94)	128 (91)	77 (91)	0.305	290 (90)	219 (96)	61 (97)	0.002 *
	Neutral	18 (7)	15 (5)		6 (6)	27 (5)		17 (4)	11 (8)	5 (6)		29 (9)	3 (1)	1 (2)	
	Disagree	4 (3)	7 (1)		2 (2)	9 (2)		7 (2)	1 (1)	3 (4)		5 (2)	5 (2)	1 (2)	
There is not much that people like me can do to stop antibiotic resistance	Agree	153 (49)	108 (39)	0.031 *	59 (56)	198 (40)	<0.001 *	161 (41)	61 (44)	39 (46)	0.293	142 (44)	92 (41)	27 (43)	0.012 *
	Neutral	119 (27)	59 (30)		33 (31)	142 (28)		117 (30)	33 (24)	28 (33)		108 (33)	54 (24)	16 (25)	
	Disagree	123 (24)	52 (31)		14 (13)	160 (32)		111 (29)	46 (33)	18 (21)		74 (23)	81 (36)	20 (32)	
I am worried about the impact that antibiotic resistance will have on my health, and that of my family	Agree	307 (74)	162 (78)	0.351	77 (73)	386 (77)	0.604	303 (78)	99 (71)	67 (79)	0.423	234 (72)	187 (82)	48 (76)	0.025 *
	Neutral	75 (21)	45 (19)		24 (23)	94 (19)		71 (18)	33 (24)	16 (19)		73 (23)	32 (14)	15 (24)	
	Disagree	13 (5)	12 (3)		5 (5)	20 (4)		15 (4)	8 (6)	2 (2)		17 (5)	8 (4)	0 (0)	
As long as I take antibiotics correctly, antibiotic resistance is not a problem for me	Agree	193 (45)	99 (49)	0.409	64 (60)	225 (45)	0.003 *	178 (46)	65 (46)	49 (58)	0.190	166 (51)	95 (42)	31 (49)	0.007 *
	Neutral	81 (25)	55 (21)		24 (23)	111 (22)		83 (21)	35 (25)	18 (21)		76 (23)	43 (19)	17 (27)	
	Disagree	121 (30)	65 (31)		18 (17)	164 (33)		128 (33)	40 (29)	18 (21)		82 (25)	89 (39)	15 (24)	

Data presented as *n* (%). * *p*-value < 0.05.

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
