# Peer review of "Antibiotic Use and Antibiotic Resistance: Public Awareness Survey in the Republic of Cyprus"

_antibiotics, 2020, doi:10.3390/antibiotics9110759_

Round 1

Reviewer 1 Report

In previous reports Cyprus was secribed as having lower level of public knowledge about antibiotic use and antibiotic resistance compared to other European countries, however improvements were also seen in the past years. To identify the areas for future intervention programes, it was a good idea to use the more deatiled WHO questionnaire in this survey, despite the existing results of the Eurobarometer 478 of the same country.

However, there are some methodological issues that makes the manuscript a lower quality. Sampling and representativeness of the recent publication is doubtful. Selection of population is not clear and specified (just saying that it was done in public areas and by social networks is not enough), it projects selection bias and consequently lack of representativeness. Performance bias can be identified as well, since questionnaire completion was done on a mixed way. Sample sizes are not proportional the population size of each districts of the country. Some sample sizes were too small to use for analysis. Age-, gender or educational distribution also do not represent the country or districts. Moreover, not all districts of Cyprus presented data. There was no detailed information about minimum sample size calculation, response rate, percentage of incomplete questionnaires, or questionnaire exclusion. In data processing it would have been useful to classify and analyse some answers together (e.g. agree and slightly agree).

Table 2 and 3 should have been described in the text in more details. Overall the intrepretation of the results is superficial. Some section of the results lacks the correlation with some (household composistion) or all sociodemographic parameters (perception of respondents towards antibiotic use and antibiotic resistance).

As representativeness is questionable it was too strong to draw generalized conclusion, although the authors themself stated somehow in the coclusison that some issues affected the representation.

As the journal has no length restriction, it would have been good to interpret the research in a deeper way, in more details. So as to include more references (not just 19) as this is an extensively published topic in literature.

Author Response

Reviewer 1

Comments and Suggestions for Authors

In previous reports Cyprus was secribed as having lower level of public knowledge about antibiotic use and antibiotic resistance compared to other European countries, however improvements were also seen in the past years. To identify the areas for future intervention programes, it was a good idea to use the more deatiled WHO questionnaire in this survey, despite the existing results of the Eurobarometer 478 of the same country.

Thank you for your positive comments. We agree that an indepth survey was needed to detect specific targets for intervention.

However, there are some methodological issues that makes the manuscript a lower quality.

Sampling and representativeness of the recent publication is doubtful. Selection of population is not clear and specified (just saying that it was done in public areas and by social networks is not enough), it projects selection bias and consequently lack of representativeness. Performance bias can be identified as well, since questionnaire completion was done on a mixed way. Sample sizes are not proportional the population size of each districts of the country. Some sample sizes were too small to use for analysis. Age-, gender or educational distribution also do not represent the country or districts. Moreover, not all districts of Cyprus presented data.

Reviewer is correct and we have modified our methods to address the selection of population and updated our limitations paragraph accordingly to acknowledge these drawbacks. We agree that bias in responses may be owed to variable representativeness owed to the characteristics of our respondents and have therefore updated our discussion to address these issues. Allow us to note that responses from all 5 districts of Cyprus (government-controlled areas) were included. Although distribution was not universally proportional to the population of all districts, representation was not significantly different, with a slightly higher proportion of responses from Nicosia and lower from Limassol compared to the population of each district. This comment is now added in our limitations paragraph.

There was no detailed information about minimum sample size calculation, response rate, percentage of incomplete questionnaires, or questionnaire exclusion.

We have added relevant details in the Methods section (section 4.2). Sample size calculation was not made, as the methodology was based on the WHO study, which adopted the methodology of previous surveys of the WHO and the European Commission, as described in the 2.2 Section of the “Antibiotic resistance: multi-country public awareness survey” of the WHO. Of note, compared to the population of nearly all countries included in the previous surveys that followed the same WHO methodology (Sri Lanka, Australia, Mexico, Nigeria, Italy, Russian Federation, Egypt, Sudan, Serbia, China, Vietnam, Indonesia, India, South Africa), the sample size of our survey yielded a much higher rate of respondents per population. If the reviewer deems necessary, we can also mention this in our discussion.

In data processing it would have been useful to classify and analyse some answers together (e.g. agree and slightly agree).

We thank the reviewer for the suggestion. We updated the manuscript and all analyses are presented with responses grouped (Agree/Slightly agree, Neutral, Disagree/Slightly disagree)  in Figures 2 & 3 and Table 3.

Table 2 and 3 should have been described in the text in more details. Overall the intrepretation of the results is superficial. Some section of the results lacks the correlation with some (household composistion) or all sociodemographic parameters (perception of respondents towards antibiotic use and antibiotic resistance).

We agree with the reviewer that some data needed further description. We incorporated these suggestions in the revised manuscript in both Methods and Results sections. Specifically, we added 2 figures (Figure 2 & 3) as well as Table 3. Results are revised accordingly..

As representativeness is questionable it was too strong to draw generalized conclusion, although the authors themself stated somehow in the coclusison that some issues affected the representation.

Thank you  for acknowledging this. As mentioned above, we have expanded our limitations section in order to be more clear in addressing our study drawbacks.

As the journal has no length restriction, it would have been good to interpret the research in a deeper way, in more details. So as to include more references (not just 19) as this is an extensively published topic in literature.

Thank you for this comment, we agree that representativeness should be interpreted with caution and following reviewer’s advice, we have emphasized this in a new limitations paragraph. We have also expanded our discussion by further discussing our findings and adding relevant literature.

Reviewer 2 Report

This manuscript reports a public survey assessing antibiotic knowledge in a Cypriot population.

The introduction could better clarify the relationship between the current survey and existing (EU) surveys. It is not clear whether this survey is a duplication of previous surveys but done in a new country or whether the current survey uses previous surveys and expands on some questions. The methods indicate it is exactly the same survey as done in 2015 but the 12 original countries didn't include Cyprus?

How the study was advertised to participants is not explicit. it is not clear how authors aimed to get a representative sample. it sounds like a convenience sample? it appears some participants filled out the survey on paper so they were approached by a researcher in person? More detail needs to be given.

The percentage of university graduates in the sample seems high indicating the sample may not be representative?

Under results 2.3, I'm not sure I would agree with the statement that all UTIs need antibiotics but maybe this reflects the guidance in Cyprus? Also stating that all sore throats do not need antibiotics doesn't ring true entirely? I'm not sure the wording of the question (if it is as it appears in Figure 1) is entirely clear.

The discussion mentions some survey items which do not appear to be presented in the results - line 128-132?

Im not sure I am convinced by the argument in the discussion that the sample is representative for reasons discussed above.

Overall the survey highlights areas where public knowledge of antibiotics could be improved but the authors do not specify how this should be used to inform future public campaigns of interventions. Of the knowledge that was tested it is necessary for the public to score high on all of these things or are some of them more important than others?

Author Response

Reviewer 2

Comments and Suggestions for Authors

This manuscript reports a public survey assessing antibiotic knowledge in a Cypriot population.

The introduction could better clarify the relationship between the current survey and existing (EU) surveys. It is not clear whether this survey is a duplication of previous surveys but done in a new country or whether the current survey uses previous surveys and expands on some questions. The methods indicate it is exactly the same survey as done in 2015 but the 12 original countries didn't include Cyprus?

The introduction has been expanded accordingly. Indeed, our survey adopted the methodology and questionnaire of the WHO “Antibiotic resistance: multi-country public awareness survey”, which was initially performed in 12 countries . Following permission, the survey was susequently performed in other countries, including Cyprus (ie.our study) and Italy, Nigeria, Australia, Sri Lanka, Brunei (please see sections 4.1 and 4.2 of our Methods where we now describe this survey in detail).

How the study was advertised to participants is not explicit. it is not clear how authors aimed to get a representative sample. it sounds like a convenience sample? it appears some participants filled out the survey on paper so they were approached by a researcher in person? More detail needs to be given.

We have expanded our Methods section (section 4.2) to further clarify our sampling method. Furthermore, we have modified our limitations paragraph to address relevant limitations.

The percentage of university graduates in the sample seems high indicating the sample may not be representative?

We agree, although Cyprus is among the countries with the highest percentage of high education attainment in the EU. To this end, please see our limitations paragraph where we discuss this: Geographic distribution could in turn have affected representation of the education level of participants, as most respondents were higher education graduates (80.1%). Admittedly, the latter finding is higher compared to the current proportion of higher education attainment in Cyprus, albeit among the highest in Europe (49.5% for those aged 25-54 years and 23.5% for those aged 55+ years in 2019) [29]. This further enforces the necessity of proposed interventions, as we have shown association between education level and knowledge and understanding.”

Under results 2.3, I'm not sure I would agree with the statement that all UTIs need antibiotics but maybe this reflects the guidance in Cyprus? Also stating that all sore throats do not need antibiotics doesn't ring true entirely? I'm not sure the wording of the question (if it is as it appears in Figure 1) is entirely clear.

Although the reviewer has a point, these statements were taken directly from the WHO Survey questionnaire. To this end and in accordance with the WHO statements, we now clarify in the Figure 1 legend that these are conditions that “can” be treated with antibiotics and have modified the corresponding excerpts in the text.

The discussion mentions some survey items which do not appear to be presented in the results - line 128-132?

Please allow us to clarify that these items are described in Section 2.3 of the results: “...and 79.6% stated that it is not acceptable to use the same antibiotics taken from a family member or friend with the same symptoms.

Misconception was recorded when respondents were shown the statement “It’s acceptable to buy the same antibiotics, or request these from a doctor, if you’re sick and they helped you get better when you had the same symptoms before”, where 25.4% incorrectly agreed.”

Im not sure I am convinced by the argument in the discussion that the sample is representative for reasons discussed above.

Thank you for this comment. We have updated our limitations section to address issues arising from sampling and representativeness. “As for population representativeness, compared to the previous studies that followed the WHO methodology, the number of respondents is considered far more representative of the total population of the Republic of Cyprus (70 respondents /100.000 population) [19,21,22,26,28]. There was sufficient participation from different age groups, with those in young adulthood (25-44 years of age) representing 65% of the population surveyed and corresponding well to the age distribution of the Cyprus population [28].  The responses of the current survey were gathered from a convenience sample, including results from both online and face-to-face responses, might have led to response bias, where people answer questions on the basis of what is “expected” of them to answer. However, approximately 50% completed the survey during face-to-face interviews and therefore this type of bias was reduced. The survey also contained several statements asking the respondents whether they agreed. This could lead to acquiescence bias, where respondents are more likely to agree rather than disagree, regardless of the content. Nevertheless, this survey followed an established and validated methodology by the WHO, with important findings from a notable sample size of the general public of Cyprus, a country characterised by high antimicrobial use and alarming antimicrobial resistance rates.”

Overall the survey highlights areas where public knowledge of antibiotics could be improved but the authors do not specify how this should be used to inform future public campaigns of interventions. Of the knowledge that was tested it is necessary for the public to score high on all of these things or are some of them more important than others?

Reviewer is very accurate in their comment. Although there was no specific score required, we analyzed each question separately to detect gaps, in order to use these as future targets. Indeed, as detailed in the results and discussed, such points were recorded.

Reviewer 3 Report

Overall, this article on the knowledge and understanding of antibiotic use and resistance of the population of Cyprus was a straightforward article that may be useful in informing future efforts to improve antibiotic use in Cyprus. Below are some suggestions for improvement.

Broad comments:

  • Since one of the socio-demographic characteristics of the participants is “received a prescription for last antibiotic use,” I think it would be helpful to clarify on ways people can get an antibiotic if it has not been prescribed for them. Is it legal in Cyprus for people to purchase antibiotics from pharmacies without a prescription?
  • If available, including more data on the increased prescribing in Cyprus compared to other countries would be helpful. If there is data available on the inappropriate prescribing patterns, that could be helpful to include.
  • Since most people had higher levels of education, I would include some more background on the Cyprus population. Does the majority of the population have a higher level? If not, is this a misrepresentation?
  • Would the following article be useful to include in the discussion? Rousounidis A, et al. Descriptive study on parents' knowledge, attitudes and practices on antibiotic use and misuse in children with upper respiratory tract infections in Cyprus. Int J Environ Res Public Health. 2011 Aug;8(8):3246-62.

Specific comments:

  • Figure 1: Label the x-axis. Consider providing a more descriptive title to reflect what data is being shown rather than just the question.
  • Table 2: I’m not sure why some of the numbers have “/” after them.
  • Line 114-115: Needs citation.
  • Line 123-125: Please clarify on specific targets mentioned and if any were implemented due to this study.
  • Line 126: WHO should be defined the first time it is used. Will not need to be defined on line 182 if added here.
  • Line 126-127: Needs citation.
  • Line 164-165: How did you gather that the number of respondents was considered representative?
  • Include an English translation of the supplementary materials. If this is available somewhere on the WHO website, please provide a citation and direct link so that international readers can understand it.
  • Line 200: You mention the survey was done in public areas. Please expand on what constitutes a public area. Would this explain why the majority of participants were from the capital?
  • For citation 10, provide a link/URL to the survey results in the reference section.

Author Response

Reviewer 3

Comments and Suggestions for Authors

Overall, this article on the knowledge and understanding of antibiotic use and resistance of the population of Cyprus was a straightforward article that may be useful in informing future efforts to improve antibiotic use in Cyprus. Below are some suggestions for improvement.

Thank you for your positive remarks and acknowledging the importance of our findings.

Broad comments:

  • Since one of the socio-demographic characteristics of the participants is “received a prescription for last antibiotic use,” I think it would be helpful to clarify on ways people can get an antibiotic if it has not been prescribed for them. Is it legal in Cyprus for people to purchase antibiotics from pharmacies without a prescription?

Since the establishment of the General Health System in Cyprus (June 2019), antibiotics are given only through electronic prescription and over-the-counter antibiotics are not allowed. We have added this in our discussion. “Since the establishment of the General Health System in Cyprus (June 2019), antibiotics are given only through electronic prescription and over-the-counter antibiotics are not allowed. We now comment on this in our discussion. “The present survey was performed 6 months after the establishment of the General Health System in Cyprus, which mandates antibiotic use only through electronic prescription and over-the-counter antibiotics are not allowed. Although 50% of participants reported having received their last antibiotic course at least a year ago (Table 1), 87% reported using a prescription. Therefore, the proportion of respondents that could have used antibiotics over-the-counter is low.”

  • If available, including more data on the increased prescribing in Cyprus compared to other countries would be helpful. If there is data available on the inappropriate prescribing patterns, that could be helpful to include.

Although such information is very limited, we have expanded information in the Introduction in relation to the high antibiotic use and examples of prescribing patterns. “In Cyprus, results from local surveillance of antimicrobial use and resistance are limited to international studies. According to recent surveillance reports [8], Cyprus is among the highest consumers of antibiotics in the community and hospitals in Europe, with extended spectrum antibiotics (eg.third-fourth generation cephalosporins, fluoroquinolones, carbapenems) accounting for more than 50% of antibiotic use in hospitals [9]. As high as 95% of surgical prophylaxis is administerd for more than 1 day in Cyprus, reflecting the degree of inappropriate antibiotic use. Antibiotic-resistant bacteria are isolated in up to 51.4% of healthcare-associated infections in Cyprus, which is exceeded by only 6 countries in the European region and is alarmingly high compared to the European average of 31.6% [10].  In addition, previous surveys (eg.the Eurobarometer survey) have shown that Cyprus citizens have exhibited certain knowledge gaps in regards to indications for antibiotic treatment, such as the common cold and the flu [5]. This is further supported by a previous survey among parents in Cyprus, where lower education level was associated with antibiotic misuse in their children [11].

  • Since most people had higher levels of education, I would include some more background on the Cyprus population. Does the majority of the population have a higher level? If not, is this a misrepresentation?

This is an accurate comment; we have further elaborated on the education level of the Cyprus population in relation to our findings in the limitations paragraph: “Geographic distribution could in turn have affected representation of the education level of participants, as most respondents were higher education graduates (80.1%). Admittedly, the latter finding is higher compared to the current proportion of higher education attainment in Cyprus, albeit among the highest in Europe (49.5% for those aged 25-54 years and 23.5% for those aged 55+ years in 2019) [29]. This further enforces the necessity of proposed interventions, as we have shown association between education level and knowledge and understanding.”

  • Would the following article be useful to include in the discussion? Rousounidis A, et al. Descriptive study on parents' knowledge, attitudes and practices on antibiotic use and misuse in children with upper respiratory tract infections in Cyprus. Int J Environ Res Public Health. 2011 Aug;8(8):3246-62.

Thank you for this useful suggestion; we have now added pertinent information from this survey in our Introduction and Discussion.

Specific comments:

  • Figure 1: Label the x-axis. Consider providing a more descriptive title to reflect what data is being shown rather than just the question.

Thank you for noting this. We have now modified Figure legend and clarified the x-axis.

  • Table 2: I’m not sure why some of the numbers have “/” after them.

We apologize for these typos, which we have now corrected.

  • Line 114-115: Needs citation.

Added

  •  
  • Line 123-125: Please clarify on specific targets mentioned and if any were implemented due to this study.

We have now expanded on this part and have added details to mention specific targets. None were implemented yet, as this survey was the first of its kind. However, following reviewer’s suggestion, we have proceeded to elaborate on ways to address these issues, based on our results and existing literature. A characteristic example is the first part of our Discussion: “Recent surveillance reports highlight that Cyprus has a high prevalence of antibiotic-resistant bacteria, as well as a high overall consumption of antibiotics [9,10,12]. The current survey provides essential information on knowledge and understanding of the public of Cyprus on use of antibiotics and antibiotic resistance and delineates areas for future interventions to improve antibiotic use and decrease inappropriate use. Specifically, the results of this survey show that even though the general public of Cyprus have basic knowledge and exhibit awareness on appropriate antibiotic use and antibiotic resistance, they do not fully understand basic premises, such as the causes of antibiotic resistance or their own role in decreasing it. Certain knowledge gaps and misconceptions were detected regarding mechanisms leading to antibiotic resistance, which diseases are indicated for antibiotic treatment and the need for a multifaceted approach to tackle resistance [13]. Consequently, specific targets are set for improvement of knowledge and awareness on the mechanisms leading to antibiotic resistance, as well as which diseases can be treated with antibiotics and which cannot. Moreover, our study has highlighted the importance of enabling public understanding of their own role in reducing antibiotic resistance and the need for a multidisciplinary approach to tackle resistance [14].

Optimization of antibiotic use in healthcare requires a national strategy with specific priorities and with participation of different stakeholders involved in antibiotic use, such as prescribers, pharmacists, patients and healthcare organizations. A minimum set of knowledge and competences in microbiology, infection pathogenesis, prescribing and stewardship, is necessary for prescribers in order to ensure safe and responsible antibiotic use [15]. In addition to organised antimicrobial stewardship and surveillance programs, interventions that have shown to improve antibiotic use-related indices have included setting standards and quality indicators for antibiotic use, and supporting doctor-patient communication [16]. Especially for the latter, training physicians in communication skills and encouraging shared decision making when prescribing antibiotics to a patient, have been repeatedly shown to improve the quality of antibiotic use without affecting outcomes [17,18].”

  • Line 126: WHO should be defined the first time it is used. Will not need to be defined on line 182 if added here.

Corrected

  • Line 126-127: Needs citation.

Added

  • Line 164-165: How did you gather that the number of respondents was considered representative?

We adopted the methodology of the WHO study, which followed the methodology of previous surveys of the WHO and the European Commision, as described in the 2.2 Section of the “Antibiotic resistance: multi-country public awareness survey” of the WHO, where a sample size of 1000 per country was sought where an online methodology was adopted and a sample of 500 per country where it was necessary to use face-to-face. Of note, compared to the population of nearly all countries included in the previous surveys that followed the same WHO methodology (Sri Lanka, Australia, Mexico, Nigeria, Italy, Russian Federation, Egypt, Sudan, Serbia, China, Vietnam, Indonesia, India, South Africa), the sample size of our survey yielded a much higher rate of respondents per population. We have added this in our limitations paragraph in the Discussion: “As for population representativeness, compared to the previous studies that followed the WHO methodology, the number of respondents is considered far more representative of the total population of the Republic of Cyprus (70 respondents /100.000 population) [19,21,22,26,28]. There was sufficient participation from different age groups, with those in young adulthood (25-44 years of age) representing 65% of the population surveyed and corresponding well to the age distribution of the Cyprus population [28].”

  • Include an English translation of the supplementary materials. If this is available somewhere on the WHO website, please provide a citation and direct link so that international readers can understand it.

This is now added in Section 4.1 of Methods. “The questionnaire and detailed study methodology area available at https://apps.who.int/iris/handle/10665/194460.”

  • Line 200: You mention the survey was done in public areas. Please expand on what constitutes a public area. Would this explain why the majority of participants were from the capital?

We now describe public areas and how responses were sought, in Section 4.2 of Methods. Indeed, this could explain that 49% of participants were from the capital (Nicosia), although Nicosia district accounts for 40% of the total population of Cyprus.

  • For citation 10, provide a link/URL to the survey results in the reference section.

Corrected

Reviewer 4 Report

Mikaela Michaelidou et al performed a cross sectional survey aiming to assess the current knowledge on antimicrobial resistance in the population of Cyprus.

The authors adapted a well know tool developed by WHO for this purposes.

Manuscript is of interest and provides a great background for future public health interventions. References are up to date.

Education and living area play some role in the basic understanding of such a problem.

Prior antibiotic use: 13% of responses had either not received or remembered a prescription for their last antibiotic course; Is Cyprus a country where antibiotics can be bought over the counter? If so, please clarify.

Did authors capture number of antibiotic courses in addition to timing from last antibiotic course?

Did prior antibiotic use had any effect in basic knowledge of the topic? One could hypothesize that those who receive multiple courses or had a prescription last month (as compared to over a year) might have better understanding of the problem.

Can you add or explain if there was any difference in knowledge between those homes with children vs those without. One could think those caring for children might be more aware of the problem.

Author Response

Reviewer 4

Comments and Suggestions for Authors

Mikaela Michaelidou et al performed a cross sectional survey aiming to assess the current knowledge on antimicrobial resistance in the population of Cyprus.

The authors adapted a well know tool developed by WHO for this purposes.

Manuscript is of interest and provides a great background for future public health interventions. References are up to date.

Education and living area play some role in the basic understanding of such a problem.

Thank you for your positive remarks and for highlighting the importance of our study in guiding future public health interventions.

Prior antibiotic use: 13% of responses had either not received or remembered a prescription for their last antibiotic course; Is Cyprus a country where antibiotics can be bought over the counter? If so, please clarify.

Since the establishment of the General Health System in Cyprus (June 2019), antibiotics are given only through electronic prescription and over-the-counter antibiotics are not allowed. We now comment on this in our discussion. “The present survey was performed 6 months after the establishment of the General Health System in Cyprus, which mandates antibiotic use only through electronic prescription and over-the-counter antibiotics are not allowed. Although 50% of participants reported having received their last antibiotic course at least a year ago (Table 1), 87% reported using a prescription. Therefore, the proportion of respondents that could have used antibiotics over-the-counter is low.”

Did authors capture number of antibiotic courses in addition to timing from last antibiotic course?

This is a valid point. We’re afraid that the WHO questionnaire used in this survey did not seek this information and focused on the last antibiotic course of the respondents.

Did prior antibiotic use had any effect in basic knowledge of the topic? One could hypothesize that those who receive multiple courses or had a prescription last month (as compared to over a year) might have better understanding of the problem.

Thank you for the suggestion. Following reviewer suggestion, we added information from this analysis. However, no significant differences worth noting were detected between those that received antibiotics <1 year vs >1 year ago.

Can you add or explain if there was any difference in knowledge between those homes with children vs those without. One could think those caring for children might be more aware of the problem.

We thank the reviewer for the suggestion. We have performed the suggested analyses and have incorporated a new Table (Table 3).

Round 2

Reviewer 1 Report

Thank you very much for reworking the article, substantial quality imrovement can be seen. I accept all the updates and corrections.

It is very appreciated that you contracted the answers in the analysis for figure 2 and 3 and table 3, but you must indicate this in the methodologocal section.

In the new version Figure 2 and 3 is empty (it must be a visualization problem), therefore I can not judge them in this way, but I believe that theay are ok.

Reviewer 2 Report

The authors have addressed my previous comments and I believe the manuscript has improved substantially and now provides more detail about how the study was done and the results themselves.